# Physical, Mechanical, and Morphological Performances of Arrowroot (*Maranta arundinacea*) Fiber Reinforced Arrowroot Starch Biopolymer Composites

**DOI:** 10.3390/polym14030388

**Published:** 2022-01-19

**Authors:** J. Tarique, E. S. Zainudin, S. M. Sapuan, R. A. Ilyas, A. Khalina

**Affiliations:** 1Advanced Engineering Materials and Composites Research Centre (AEMC), Department of Mechanical and Manufacturing Engineering, Universiti Putra Malaysia, Serdang 43400, Malaysia; tarique5496@gmail.com (J.T.); sapuan@gmail.com (S.M.S.); 2Laboratory of Biocomposite Technology, Institute of Tropical Forest and Forest Products (INTROP), Universiti Putra Malaysia, Serdang 43400, Malaysia; khalina@upm.edu.my; 3School of Chemical and Energy Engineering, Faculty of Engineering, Universiti Teknology Malaysia, Johor Bahru 81310, Malaysia; ahmadilyas@utm.my; 4Centre for Advanced Composite Materials (CACM), Universiti Teknology Malaysia, Johor Bahru 81310, Malaysia; 5Department of Biological and Agricultural Engineering, Universiti Putra Malaysia, Serdang 43400, Malaysia

**Keywords:** arrowroot fibers, arrowroot starch, plasticizer, mechanical properties, physical properties, morphological properties

## Abstract

This research is driven by stringent environmental legislation requiring the consumption and use of environmentally friendly materials. In this context, this paper is concerned with the development and characterization of thermoplastic arrowroot starch (TPAS) based biocomposite films by incorporating arrowroot fiber (AF) (0–10%) into a glycerol plasticized matrix by using the solution casting method. Developed TPAS/AF composite films were investigated, such as physical, morphological (FESEM), tensile, and tear strength characteristics. The tensile and tear strengths of TPAS/AF composites were increased significantly from 4.77 to 15.22 MPa and 0.87 to 1.28 MPa, respectively, as compared to the control TPAS films, which were 2.42 MPa and 0.83 MPa, respectively, while elongation was significantly decreased from 25.57 to 6.21% compared to control TPAS film, which was 46.62%. The findings revealed that after the fiber was reinforced, the mechanical properties were enhanced, and the optimum filler content was 10%. Regardless of fiber loadings, the results of water absorption testing revealed that the composite films immersed in seawater and rainwater absorbed more water than distilled water. Overall, the results of this research focus on providing information on biopolymer composite film and revealing the great potential it has for the food packaging industry.

## 1. Introduction

Fossil fuel-based polymers are one of the most common materials used in the packaging industry, which has long been a source of concern for the global ecosystem. The overwhelming volume of environmentally hazardous plastic waste has prompted the research of polymers from natural sources, renewable, sustainable, and biodegradable. Biopolymers have been explored as potential alternatives for fossil-based plastics in an attempt to address the increasing environmental problems created by non-degradable plastics [1,2]. In consequence, various thermoplastic composites have been employed to make eco-friendly packaging products.

Therefore, the use of agricultural residues as sustainable filler materials has grown in significant popularity. Low cost, low density, and decreased tool wear are all recognized benefits of such resources, which benefit the manufacturing industry [3]. Various natural waste types, such as sugar palm, cassava, corn, and others, have been discovered to be suitable for usage as polymer composite reinforcements [4,5,6]. Advances in the implementation of technologies used for the development of biomaterials depend on those derived from vegetable by-product-based formulations, as well as fruit [7,8].

Solution casting is a lab-scale batch casting method that has a significant impact on the microstructure and characteristics of polymeric thin films. As a result, it is critical to recognize the effects of processing conditions and use the right processing approach. It produces thin-film products of excellent quality with improved optical and physical properties [9,10]. In contrast, the blown film is a type of extrusion method that is extensively employed to fabricate packaging film in the food or medical industry on a large scale [11,12]. The blown film has a few disadvantages, such as the thickness of a film being difficult to manage precisely, the highly complex approach, and several variables that can go wrong [13].

Biocomposites are transforming the research practices conducted in the field of material engineering since several valuable characteristics have been confirmed, including lightweight, low power consumption, biodegradability, sustainability, usability, and eco-friendly properties [3,14,15]. Along with several advantages as reinforcements, there are some drawbacks associated with these lignocellulosic fibers. Therefore, a significant number of experiments on improving the compatibility of lignocellulosic fibers and other biopolymers have been conducted [16]. This interaction is hindered by the hydrophilic nature and high polymerization of cellulosic fibers, which results in poor interfacial and mechanical adhesion [17]. There is little dimensional flexibility of these biodegradable plastics when exposed to water. Surface modifications are widely used to increase the performance of cellulosic fibers and to promote increased adhesion between the starch matrix and natural reinforcement [14,15,18,19].

The tuber of arrowroot (*Maranta arundinacea*) belongs to the Marantaceae family and consists of a significant amount of starch, fiber, and carbohydrates [20]. Many wastes of arrowroot fibers are obtained during the arrowroot starch extraction process. Arrowroot starch has excellent properties, such as digestibility and gel-forming ability, as well as the highest amylose content (40.86%) [21], competing with corn starch (28–33%), cassava starch (16–19%), wheat starch (30–32%), and potato (18–20%). Previous research has shown that the amylose content of starch influences its film-forming properties; strong and stiff films are formed by hydrogen bonding linear chains together. As a result, the high amylose content of arrowroot starch produces stronger films than other starch sources. According to literature, arrowroot rhizomes comprise 38.1% of bagasse fiber [20]. Branco et al. [22] found that the arrowroot bagasse fibers are coarser as well as longer in comparison to cassava bagasse fibers. In previous work, biopolymer films were developed from arrowroot starch employing a solution casting process with 15 to 45% (by weight) glycerol as a plasticizer [23]. The results revealed that the incorporation of 30% glycerol into the biopolymer was the most effective glycerol concentration, resulting in outstanding thermal and physical properties. Despite having adequate characteristics, the developed biopolymer has certain drawbacks, such as poor tensile and water resistance [23]. Hence, to overwhelm such limitations, arrowroot starch that is reinforced by arrowroot fiber is expected to produce better outcomes.

This research is concerned with using glycerol as a plasticizer on the characteristics of biopolymer film and arrowroot bagasse fiber (2, 4, 6, 8, and 10 wt.%) as the reinforcement. The addition of arrowroot bagasse fiber as filler agents in thermoplastic biopolymer enhanced the value of the product and improved the thermoplastic biopolymer composite’s potential to be used as a completely green packaging material. Thus far, there has been no investigation on arrowroot bagasse fiber-reinforced arrowroot starch biopolymer composites published. Therefore, this research aims to reinforce the fiber into the starch derived from arrowroot tubers to optimize the mechanical, morphological, physical, and water barrier performance of the arrowroot biopolymer composite film.

## 2. Materials and Methods

### 2.1. Materials

Arrowroot bagasse fibers were extracted from arrowroot tubers and were obtained from a local market in Kuala Lumpur, Malaysia. Bagasse fiber used in this research contains cellulose 45.97%, hemicellulose 30.18%, lignin 2.78%, ash 4.29%, and fiber sizes (<300 μm) [24]. Arrowroot starch also comprises 40.86% of amylose content [21]. The glycerol was used and purchased from Evergreen Engineering & Resources Sdn. Bhd., Semenyih, Malaysia.

### 2.2. Development of Thermoplastic Arrowroot Starch (TPAS)/Arrowroot Fiber (AF) Biopolymer Composites

Biopolymer composites were developed using the conventional solution casting method. The arrowroot starch had been dispersed in distilled water 5% (*w/w*, dry basis) to obtain the film-forming solution. The glycerol 1.5 g (*w/w*, dry starch basis) was incorporated into the solution, based on previous research. Arrowroot bagasse fiber was added to the solution as filler at (0 to 10 wt.%) according to the formulation shown in Table 1. These solutions were placed in a thermostatic bath (Daihan Scientific, Jalan Buroh, Singapore) and heated at 80 ± 3 °C under slow stirring for 15 min. Before casting, the biopolymer forming solutions were cooled down and put in a vacuum desiccator to prevent air bubbles from forming. After that, 50 g of each solution was cast in Petri dishes (Diameter: 120 mm, Evergreen Engineering & Resources Sdn. Bhd., Semenyih, Malaysia) to work as a casting surface, resulting in a high surface finish. The cast plates were kept in an airflow oven at 45 °C for 24 h for dehydration. Then, the dehydrated samples were conditioned at 25 °C for 24 h and detached from the plates, as shown in Figure 1. Before characterization, the film samples were stored at 25 ± 3 °C and 52% of relative humidity for a week.

### 2.3. Characterization of Biocomposites

#### 2.3.1. The Thickness of Biopolymer Composite Films

A micrometer (Mitutoyo-co, Kawasaki, Japan) was used to determine the thickness of biopolymer composite film with a precision of 0.001 mm. The ASTM method F2251 [25] based technique was used to determine each biopolymer film replicates, and the mean value was determined to get the actual thickness.

#### 2.3.2. Density

The film sample size of (20 mm × 20 mm) was used to determine the film density based on ASTM D792-13 [26] and Equation (1):(1)$$\rho =\frac{m}{v}$$
where *m* and *v* are the weight (g) and volume (cm^3^) of the film sample, respectively. The volume of the sample was calculated by the multiplication of film area and thickness.

#### 2.3.3. Water Contents (W_C_)

The water content of three replicates of each sample was determined using ASTM D664-07 [27]. All samples were weighed (*M_i_*) before being dried at 105 °C for 24 h and reweighed (*M_f_*). Equation (2) was used to evaluate the moisture content of each film sample.
(2)$$W_{C}\;(\%)=\left(\frac{M_{i}-M_{f}}{M_{i}}\right)\times 100$$

#### 2.3.4. Water Solubility (W_S_)

The films’ solubility was measured by the mean of the technique given by Shojaee-Aliabadi et al. [28]. Initially, the film sample size of (30 × 10 mm^2^) was dried at the temperature of 105 °C for 24 h and then weighted to obtain the initial weight (*W_i_*). After that, each specimen strip was soaked in a beaker (150 mL) of distilled water and continuously stirred at 500 rpm for 6 h at 23 ± 2 °C. The remaining part of the samples was placed to dehydrate at 105 °C for 24 h and reweighed to get the final weight (*W_f_*). Finally, Equation (3) was used to calculate the Ws (%) of the composite samples.
(3)$$W_{S}\;(\%)=\left(\frac{W_{i}-W_{f}}{W_{i}}\right)\times 100$$

#### 2.3.5. Water Absorption (W_A_)

The water absorptions were determined based on ASTM D 570-98 [29]. The samples were dehydrated at 50 °C for 24 h before being placed in a desiccator to cool them to confirm the constant weight. The specimen was then weighed (*W_0_*) and immersed in distilled water at 23 ± 2 °C. In an interval, the immersed samples were wiped down with a dry piece of cloth and reweighed (*W_i_*). Both final and initial weights were used in Equation (4) to compute the water absorption of films.
(4)$$W_{A}\;(\%)=\left(\frac{W_{i}-W_{0}}{W_{i}}\right)\times 100$$

#### 2.3.6. Fourier Transform Infrared Spectroscopy (FTIR)

FTIR spectra of the specimens were recorded using a spectrometer (Thermo Fisher Scientific, model Nicolet 6700, Rockford, IL, USA) to investigate the presence of functional groups in the biopolymer composite specimens. The analysis was carried out with 16 scans of 4000–650 cm^−1^ for each specimen, with a spectral resolution of 4 cm^−1^.

#### 2.3.7. Field Emission Scanning Electron Microscopy (FESEM)

A field emission electron microscope was used to examine the morphological behavior of biocomposite film (FEI Nova NanoSEM 230, Brno, Czech Republic). In order to avoid unwanted charging, the complete specimens were coated with gold by employing an argon plasma metalized model (sputter-coater K575X, Edwards Limited, Crawley, UK) [30]. Eventually, the FESEM experiment was carried out at a 3 kV acceleration voltage.

#### 2.3.8. Mechanical Testing

Tensile behaviors of biocomposite samples were tested at room temperature by employing a 5 kN Instron 3365 tensile machine (Instron, Norwood, MA, USA) following the ASTM D882-02 standard [31]. The sample (70 × 10 mm^2^) was fixed properly between tensile clamps. Initially, the gauge length of the sample and the machine crosshead speed were set at 30 mm and 2 mm/min, respectively. Tensile strength, tensile modulus, and elongation were measured for ten replicates of each specimen. The mechanical properties were evaluated using the mean value of the measurements.

The ASTM D-1938-8 [32] standard method was used to assess the tear propagation strength of the film samples, with some modification. Specimens (75 × 25 mm^2^) with a 12.5 mm slit were prepared according to the standard test technique. The tensile test was carried out by pulling the specimens in the Instron 3365, USA, Universal Testing Machine (UTM), with a crosshead speed of 250 mm/min and 50 mm spacing between the grips. The tests were performed at room temperature. To compare this mechanical characteristic, the tear strength of the biocomposite was normalized based on film thickness. The UTM was also used to determine the tear strength of biocomposite film.

#### 2.3.9. Statistical Analysis

Statistical analysis was analyzed using the SPSS software package. The analysis of variance (ANOVA) (IBM SPSS Statistic version 23, Armonk, NY, USA) was used to determine the possibility of significant differences (*p* ≤ 0.05) among the data. The covariance matrix has been used in MINITAB statistical software (version 18.0, Minitab Inc., State College, PA, USA) to perform the principal component analysis (PCA), a multivariate exploratory technique (State College, PA, USA). The physical and mechanical properties of the films were used as active variables in the principal component derivation, and the samples (with different fiber loadings) were projected onto the factor space.

## 3. Results and Discussion

### 3.1. Thicknesses and Densities of Biopolymer Composite Film

Based on Table 2, the thickness of AS composite films increased significantly (*p* < 0.05) with the introduction of the AF concentration, and the density also reduced significantly. The content of the dry mass per unit area of the film-forming solutions was strictly controlled during the casting process, resulting in different thickness values for controlling TPAS and TPAS/AF biocomposite films. The composite film sample AF 10% showed the highest thickness of 203 μm, compared to the other concentrations of 2%, 4% and 6%. Consequently, the composite sample AF 10% demonstrated the lowest density. The results were related to the intermolecular interaction of fibers with polymer matrix. The higher the AF concentration, the more porous the structure and lower the density compared to biopolymer film, resulting in a thicker and coarser film [33]. Similar findings were reported on the influence of fiber additions on the thicknesses and densities of biopolymer composite films [6], the biocomposite films made from sugar palm cellulose fibers and starch [34], as well as seaweed fiber-reinforced sugar palm starch-based biocomposites [35]. Nevertheless, the lower densities of composite samples made them desirable materials, particularly for applications that require compact and easy handling.

### 3.2. Water Contents (Wc)

Despite its hydrophilic nature, AF lowered moisture retention in TPAS/AF composite films, as tabulated in Table 2. In this study, the water contents of TPAS/AF films were significantly (*p* < 0.05) enhanced from (9.77 to 12.71%) by increasing fiber loadings from (2 to 10%). The composite film sample containing 10% fiber showed comparably high-water content than the control film. A small increase in the water content of TPAS/AF biocomposite films could be attributed to the low water content of AF compared to AS. These results were consistent with the results of Soykeabkaew et al. [36], where they studied water content in cassava starch foams comprising flax and jute, and Kaisangsri et al. [37], who evaluated water contents of natural fiber-reinforced/chitosan-reinforced tapioca starch biocomposites. In general, the water content increased as porosity decreased, consistent with the findings of Thymi et al. [38], and Luo et al. [39], who investigated corn starch and red lentil protein-fiber extrude, respectively.

### 3.3. Water Solubility (Ws)

Solubility of biocomposite film in water is very important in the packaging industry. The shelf life and the qualities of food packaging are maintained with lower solubility [40]. In general, a higher solubility implies a reduced resistance to water. From Table 2, it is shown that the solubility of TPAS/AF biocomposite was inferior to control biopolymer film. In the specimen of biocomposites, the initial content of AF (i.e., 2 to 10%) resulted in high water solubility; however, successive increments of fiber resulted in a drop in solubility. The TPAS/AF-10 demonstrated the lowest solubility of 22.56%, indicating that the films possessed good water stability. The ability of the fiber to interact with starch chains was due to the interaction of hydrogen bonds in starch with the hydroxyl groups present in the AF. This occurrence could be enlightened by the role of AF in preventing composite film breakdown by developing a network that tightly kept the composites together and decreased their solubility. According to Ilyas et al. [41], these interactions contributed to increased biopolymer matrix cohesiveness and reduced water sensitivity because water molecules were unable to cleavage the strong bonds. These results are following the results of Edhirej et al. [42], where they tested cassava starch/cassava peel composites, and Ibrahim et al. [5], who studied corn starch/husk composites and found that the film with less fiber loading was more soluble compared to those with high fiber loading. Likewise, Cao et al. [43] concluded that structures of matrices, as well as the resulting competitions among reinforcement material and matrix/filler interactions, are vital in determining the impact of fiber reinforcement. Moreover, Rhim and Ng [44] found that when the starch film was reinforced with the fillers, its water solubility decreased, which could be attributed to the strong hydrogen bond formation between the hydroxyl groups of the biopolymer and the hydroxyl groups of the biofiller. Dularia et al. [45] also confirmed these findings in their research investigation of biocomposites. Furthermore, while analyzing the reinforcing impact of natural fillers, Jha [46] found that the matrix structure and the subsequent interaction between matrix/filler and filler/filler were essential aspects to consider. Therefore, we can conclude that improving the water-resistant characteristics for biocomposites and in bio packaging material is crucial for prolonging the life of the product.

### 3.4. Water Absorption (W_A_)

In the packaging industry, the investigation of the water absorption behavior of biopolymer reinforced natural fiber composite is vital. Fibers in composites can absorb atmospheric moisture and water from external sources, such as rain, which can affect the performances of biocomposites for different usage. Hence, the water absorption studies of TPAS/AF biocomposite films are important for the packaging applications. The preservation of foodstuffs necessitates the use of packaging materials with considerable water resistance. A water absorption test is widely performed to determine how much water a material absorbs over time. According to Vilay et al. [47], the fiber content, temperature, permeability, orientation, and exposed area all impact water absorption analysis.

Table 3 presents the results of water absorption analysis for arrowroot fiber biocomposite films with different environmental conditions, such as distilled water, rainwater, and seawater with varying times of immersion (30, 60, and 180 min). Following Fick’s law, the weight gain caused by water absorption of the composite film with the time of immersion became more constant after 60 min [1,48]. Nevertheless, as fiber loading increased, the TPAS/AF exhibited a different propensity to absorb water. In this research, the control TPAS film displayed high water absorption because of its excellent hydrophilicity. At 30 min, the control TPAS and TPAS/AF-2 films demonstrated the highest water absorption (166% and 134%, respectively), while the TPAS/AF-8 and TPAS/AF-10 films had the lowest water absorption (90% and 91%, respectively). Despite this, water absorption values were significantly (*p* < 0.05) reduced after the arrowroot fiber was added. Consequently, increasing the fiber loading on thermoplastic composite reduced the hydrophilicity of TPAS/AF composite films. The water absorption of control films at 60 min was reduced by 47% due to the presence of fiber and their increased concentration from 2 to 10%. It was demonstrated that the control AS film sample was entirely responsible for the high-water absorption, while TPAS/AF film samples were modestly resistant to absorbing the water due to the strong interface of AF and matrix. Furthermore, the strong hydrogen bonding interactions within the thermoplastic composite film have the potential to sustain the matrix when exposed to an extremely humid environment [1].

Irrespective of fiber loading, composite films immersed in sea and rainwater absorbed more water than those immersed in distilled water, as revealed in Figure 2A–C. The seawater has a higher density (1.024 g/cm^2^) compared to rainwater (1.015 g/cm^2^) and distilled water (1.000 g/cm^2^). As a result, the samples that were immersed in seawater absorbed more water for the same volume of water absorption (weight gain) than rainwater, with TPAS/AF6, TPAS/AF8, and TPAS/AF10 biocomposites absorbing (158%, 121% and 118%) and (148%, 107% and 113%) after 3 h, respectively. Besides that, seawater has a much higher salinity (39 ppt) than rain and distilled water (0.5–3 ppt). In general, higher salinity means a greater concentration of metallic ions. The main metallic ions in seawater are K^+^, Na^+^, Mg^++^, as well as Ca^++^, which are significantly greater than rainwater. Silva et al. [49] reported that metallic particles deposit on fibers in reinforced thermoplastic composites. Due to this, it is possible to conclude that a greater concentration of metallic ions can be settled on the fiber content.

### 3.5. Fourier Transform Infrared (FTIR) Spectroscopy

The FTIR method was used to investigate the differences in structures of bio composite’s composition as well as to assess the potential reinforcement-matrix interactions. FTIR spectra of control and thermoplastic composite film samples displayed a similar structure, as shown in Figure 3 because starch and fiber were extracted from the same biological tubers. By partitioning the FTIR spectral curves into four zones, the leading bands, as well as the functional groups of the films, were examined. The first stretch zone was lower than 1500 cm^−1^, the next was between 1500 and 2800 cm^−1^, the third was the C–H stretch zone between 2800 and 3000 cm^−1^, and finally, the hydroxyl (–OH) stretch zone was considered over 3000 to 3600 cm^−1^ [50,51].

In the first zone, the presence of C–O groups within the glucose pyranose molecules of starch has caused the wide peak at 927 cm^−1^ as well as small peaks lower than 800 cm^−1^ [50]. Strong peaks at 1025 cm^−1^ as well as 1150 cm^−1^, were formed due to the vibrational bending of the C–O−H group and the linking phase of the C–C and C–O functional groups, respectively [52,53,54].

The peak at 2939 cm^−1^ could be associated with the stretching of C–H; however, a minor peak in the second zone at 1653 cm^−1^ could be attributed to the stretching of the carbonyl (C=O) group. Meanwhile, it was observed that the peak at 1653 cm^−1^ shifted to 1740 cm^−1^ (saturated aliphatic aldehydes) as adding fibers to thermoplastics. These phenomena could be attributed to the strong hydrogen bond between matrices and fillers which causes a reduction in moisture content. Besides that, the aromatic groups C–C stretching vibrations and stretching of carboxylate group at 1367 cm^−1^ wavelength, corresponded with the results of Aloui et al. [55] and Edhirej et al. [42], who used gallnut and tapioca starch-based thermoplastic biocomposites, respectively. It was observed that the peaks at 1367 cm^−1^ (C–H) were increased after fiber loadings of more than 2 wt.%. It could be attributed to the presence of cellulose in fiber [15]. A prominent peak at 1037 cm^−1^ could be linked to carbonyl functionality (C) that existed in C–O–C groups. The absorbance band between 930−1040 cm^−1^ was attributed to O–C bending in anhydroglucose rings [56].

The TPAS control film displayed a large peak in wavelength of 3600–3000 cm^−1^, which could be due to the hydroxyl (–OH) groups [48]. This was also in agreement with the work of Ilyas et al. [1], and Sahari et al. [57], who attributed the peak ranges to hydroxyl (–OH) group stretching caused by a hydrogen bond between the molecules. However, when the concentration of AF was increased, the hydrogen bond interaction between the (–OH) groups AS and AF occurred, resulting in a larger shift. For example, the associated band at 3355 cm^−1^ for control TPAS films moved to much higher positions and attained 3366 cm^−1^ after the initial fiber loadings, whereas fiber reinforcement caused it to drop significantly, indicating an enhancement in the intramolecular hydrogen bond between the starch and fiber molecule. Furthermore, the lack of new peaks demonstrates that no chemical reactions took place. These findings augmented recent findings in this field that have linked coconut fiber and cassava starch [58].

### 3.6. Morphological Properties

Figure 4 illustrates FESEM micrographs of surface and cross-sectional texture of AS-based control and TPAS/AF composite film. The control TPAS film surface demonstrated smooth surfaces with no evidence of starch granules or crack, and thus, there were no aggregates visible. This micrograph assessment was comparable to Sanyang et al. [59] in sugar palm biopolymer films. Conversely, Acosta et al. [60] investigated the morphology of starch and gelatin film and found that control films exhibited heterogeneous surface textures. Moreover, the surface of the control film (samples shown in Figure 4A) was smooth and homogeneous. However, the TPAS/AF 10% composite sample in Figure 4F was irregular. This irregularity on the surface of TPAS/AF was caused by the increasing concentration of AF and the evident agglomeration of Afs within the matrix of AS. The uneven structures showed that there was a low interfacial adhesion of AS matrices. Ibrahim et al. [5] reported a similar result, demonstrating that the film containing 2 wt.% corn husk fiber had a consistent and smooth surface and no pores in the structure. The appearance of voids was observed in the case of the addition of husk fiber from 4 to 8 wt.%, and the fracture surfaces became coarser and more rigid. Owing to the addition of AF as a filler, the cross-sectional surface of the TPAS/AF composites (2–6 wt.%) Figure 4H–J) were also rougher than the control film sample; however, they appeared much smoother compared to the respective TPAS/AF-8 and TPAS/AF-10 composite films, as shown in Figure 4K,L. This showed that lower concentrations of Afs associated more strongly with AS and distributed more evenly within the AS matrix than the higher concentration of Afs. Additionally, reinforcement of AF fibers from (4 to 8%) resulted in the formation of voids, and the fracture surfaces became rougher and stiffer. Also, the needle-like Afs were detected in cross-sections of the TPAS/AF composites, demonstrating the success of casting of biocomposites. Agglomerations of starch on the surface structure were observed. This aggregation could indicate a crosslink interaction across intramolecular and intermolecular of AS and glycerol, increasing film molecular weights [61].

In addition, the increase in compatibility between AF and AS starch was attributed to the same chemical compositions of starch and celluloses, and hydrogen bonding linkages between filler as well as the matrix. These kinds of homogeneous dispersions and adhesions of filler in the matrix played an important role in helping to improve the stiffness of TPAS/AF composite films. These findings were similar to the work of Hazrati et al. [48] and Ng et al. [62]. Surface fractures of thermoplastic cassava starch/cassava bagasse composite films were also found to be similar in appearance [6].

### 3.7. Tensile Properties

The addition of natural filler to the polymer matrix could affect the mechanical properties of the biocomposite films. Table 4 shows the analysis of variance (ANOVA) of the mechanical properties of the arrowroot starch biocomposite films. Because the *p*-value was less than 0.05 (*p* < 0.05), there was a statistically significant difference in the mean of tensile strength, elongation, Young’s modulus, and tear strength from one level of biocomposite films to the other.

Figure 5a–c show the effect of increasing the concentration of AF on the tensile properties of TPAS/AF composite films. The tensile testing was conducted to evaluate tensile strength (TS), Young’s modulus (YM), and elongation at break (EAB). The additional and increasing AFs concentration caused a decrease in EAB and significant (*p* < 0.05) increments in TS and YM, indicating that the composite films became less flexible and stiffer, and therefore more resistant [63,64]. However, when AF was added at a higher concentration of 10%, the TS and YM increased by 529%, reaching 15.22 and 1071.63 MPa, respectively, than the control film. The improvement was because cellulose and starch were compatible and structurally similar, which were from common arrowroot tubers. In addition, fibers can reduce the motion of starch molecules as well as increase the interfacial contacts between starch and fillers, resulting in a better stress transmission [65,66]. Several researchers discovered increases in TS and YM in biocomposites as fiber concentration increased [34,42,52,67,68] The effect of arrowroot fiber and arrowroot starch on the mechanical properties of starch-based biocomposite films was comparable to that of sugar palm [1] and cassava peel [42] in earlier researches. Ibrahim et al. [5] also reported an increase in TS and YM by increasing fiber concentration, reporting an increase in TS of corn starch/corn husk composite by 88% when reinforced with an 8% husk fiber. Therefore, the biocomposite films demonstrated improvements in TS and YM simultaneously. Another study by Chen et al. [69] corroborated these findings. On the contrary, the effect of fiber loading on the EAB for arrowroot starch-based composites exhibited an inverse nature when particularly in comparison to their TS and YM. This measurement aims to assess the ability of TPAS/AF biocomposite films to deform and stretch from their original length up to their point of the break. Furthermore, when control TPAS film was filled with AF, the effect of fiber loading on the TPAS/AF biocomposite film thickness and elongation at break was observed. The addition of AF from 0 to 10 wt.% increased the thickness of the biocomposite films, resulting in a decrease in elongation at the break, as shown in Table 2. The significant (*p* < 0.05) reduction in film elongation with the increase of fiber loading was attributed to intermolecular interaction, resulting in more intermolecular hydrogen bonds between the fiber and starch. By obstructing chain mobility, such reconfiguration in the starch matrix, orientation, and dispersion of the fillers stimulates the rigidity and reduced film flexibility [70].

### 3.8. Tear Strength

Tear strength is another desired strength in opening the packages when torn. In general, different biopolymers have varying tear strengths. When the packaging materials are in the full configuration, some of them may be quite difficult to tear, however, if pre-cut slits were produced, the material became an effort to propagate tears and was considerably reduced. Therefore, the initial force escalation indicates the load required to initiate the tear, after which a constant force is required to propagate it, and the longitudinal load decrease represents the final retraction of the specimen once the force is withdrawn. Figure 5d presents the load curve for control TPAS film samples with different fiber loadings but have the same total plasticizer concentration (glycerol 30%, dry starch basis). The normalized load needed to propagate tear across thermoplastic films without fibers was 0.68 N, while the load required to propagate tear across TPAS/AF-10 with 10 wt.% AF was 2 N. The presence of AF, which acts as a reinforcement agent for the TPAS matrix, is responsible for the three-fold increase. In addition, the tear strength of TPAS/AF-10 with 10% of AF was 1.28 MPa, however, it was 0.83 MPa for the control TPAS film. The tear strength of the TPAS/AF biocomposite film with 10% of AF was 1.28 MPa, although it was 0.83 MPa for the control thermoplastic film. This 1.5-fold increase was due to the influence of cellulose present in AF, which acted as a reinforcement agent in the thermoplastic matrix. Based on these findings, it is possible to conclude that the tear-propagation trend of composite films was affected by fiber reinforcement. Besides that, this mechanical property caused by AF loading was evidenced through the visual analysis of the direction where the tear propagated (Figure 6). From these studies, we found that the tear propagation trend of composite films is affected by the AF reinforcement. López et al. [71] noticed increases in tear strength when corn-based starch was filled with talc nanoparticles. Similarly, our finding agreed with the work of Wawro & Kazimierczak [72] on potato starch-based films. Additionally, it has been reported by Ismail et al. [73] that the tear strength increased from 0.521 to 1.256 MPa when the thermoplastic film was reinforced with cellulose.

### 3.9. Effect of the Fiber Loadings on the Physical and Mechanical Properties of TPAS and TPAS/AF Biocomposites: Principal Component Analysis (PCA)

The PCA was used with physical and properties data to examine the effect of the presence and concentration of fiber on the produced biocomposite films (Figure 2). The PCA plots show the similarities and differences in physical and mechanical properties between samples after fiber loading [74]. Figure 7A illustrates the PCA plot of active variables, while Figure 7B represents the thermoplastic biocomposite films evaluated based on physical and mechanical properties. PC1 described 94% of the total variance of the data, while PC2 described 5%, thus explaining 99% of the cumulative variance. Furthermore, some variables, such as WC, thickness, TS, YM, and tear strength, were found to be positively correlated with one another. While the WS, density, and EAB were all negatively correlated. The results (Figure 7A) show that the sample control TPAS was only in the third quadrant, while TPAS/AF-2 and TPAS/AF-4 were in the same (second quadrant), indicating that there was the least significant difference between these samples (there are similar). While samples TPAS/AF-6 and TPAS/AF-8 were in the first quadrant and close to samples TPAS/AF-4, indicating that these samples have the least significant difference. On the other hand, sample TPAS/AF-10 was in the fourth quadrant, indicating that there was a significant difference between these samples. In conclusion, the graph of PCA scores also revealed that the presence and concentration of the arrowroot fibers studied in the biocomposite films influenced their properties.

## 4. Conclusions

The physical, mechanical, and morphological properties of arrowroot biocomposites were investigated by using arrowroot fiber as a reinforcing agent. The test findings demonstrated that gradually increasing the fiber concentration (0 to 10%) increased the tensile performances of biocomposite with TPAS/AF films by 529%, with the TPAS/AF film having 10% AF (15.22 MPa), and tear strength increased from 0.83 MPa to 1.28 MPa, indicating effective stress transfers between matrices and fillers. FESEM micrographs displayed that the arrowroot fibers were well distributed inside the AS matrix and improved the tensile, and water absorption properties of biocomposites. The thermoplastic composite films also demonstrated better water resistance in different environmental conditions, such as rainwater, seawater, and distilled water. The lowest solubility of 22.56% was found in the TPAS/AF-10 sample, showing that the films had good water stability. The interaction of hydrogen bonds in starch with the hydroxyl groups present in the AF allowed the fiber to engage with starch chains. The improved properties of these TPAS/AF biocomposites could be attributed to excellent intermolecular hydrogen bond contacts between fibers and matrices due to their chemical resemblance, as shown by FESEM micrographs. Moreover, the information about the influence of fiber loadings on the physical and mechanical properties of the films could be helpful in the development of biopackaging films. Thus, this research demonstrated the huge potential of TPAS/AF biocomposite films for packaging applications.

## Figures and Tables

**Figure 1 polymers-14-00388-f001:**
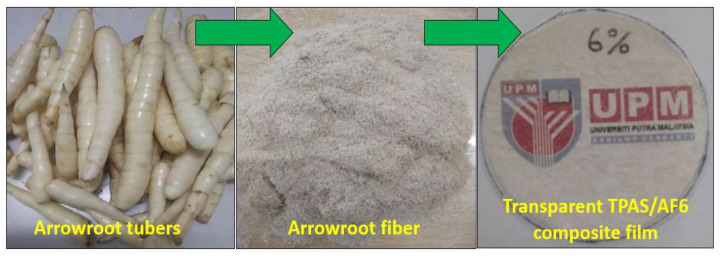
The arrowroot tubers, the fiber obtained from starch extraction, and transparent TPAS/AF composite film with 6% fiber loading.

**Figure 2 polymers-14-00388-f002:**
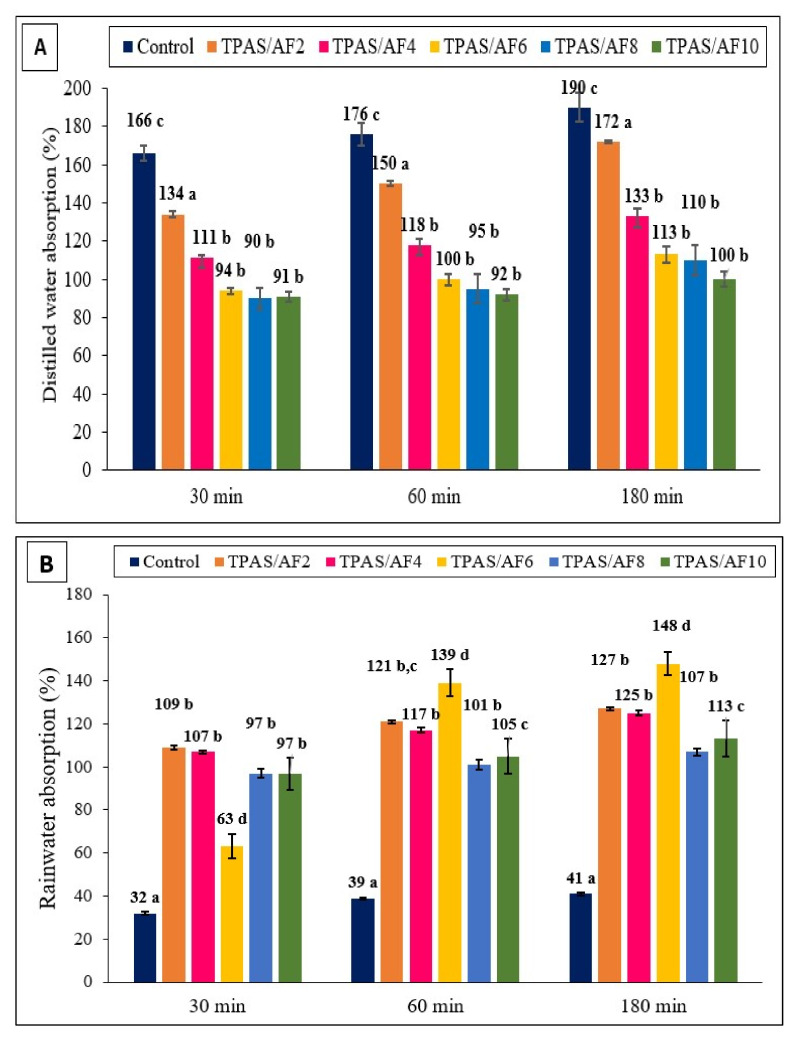
Water absorption (%) versus immersion time for the control and TPAS/AF composites in (**A**) distilled water; (**B**) rainwater; and (**C**) seawater. Values with different letters in the figures are significantly different (*p* < 0.05).

**Figure 3 polymers-14-00388-f003:**
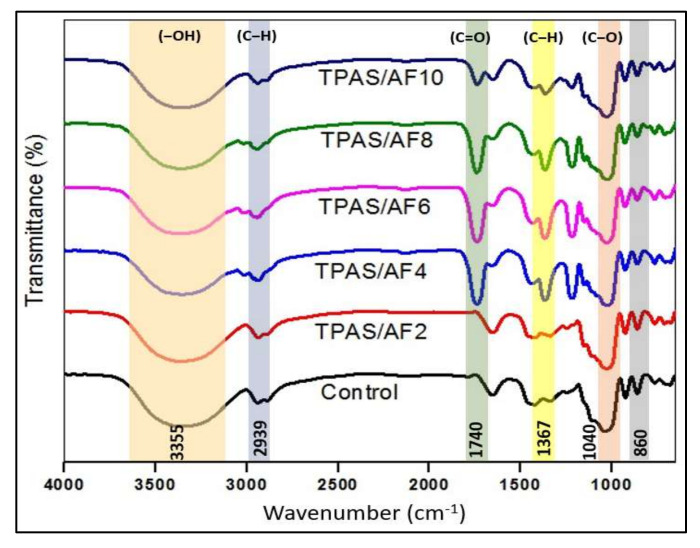
FTIR spectra of control and TPAS/AF composites with different fiber loadings.

**Figure 4 polymers-14-00388-f004:**
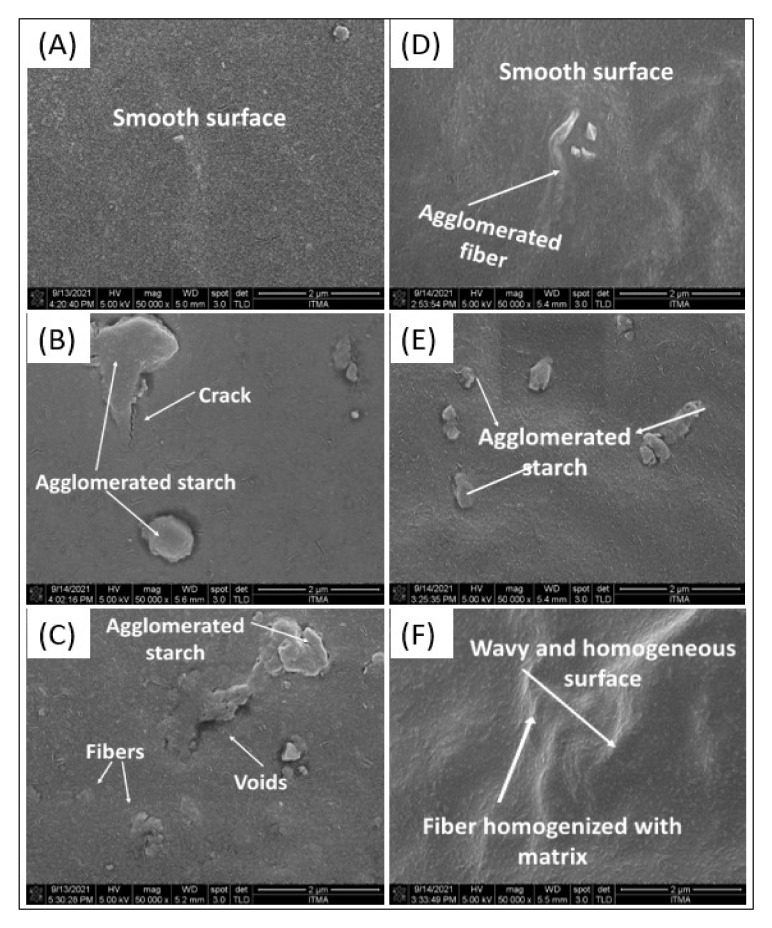
FESEM micrograph of surfaces (**A**–**F**) and cross-section (**G**–**L**) of control and TPAS/AF composites (2%, 4%, 6%, 8% and 10%), respectively.

**Figure 5 polymers-14-00388-f005:**
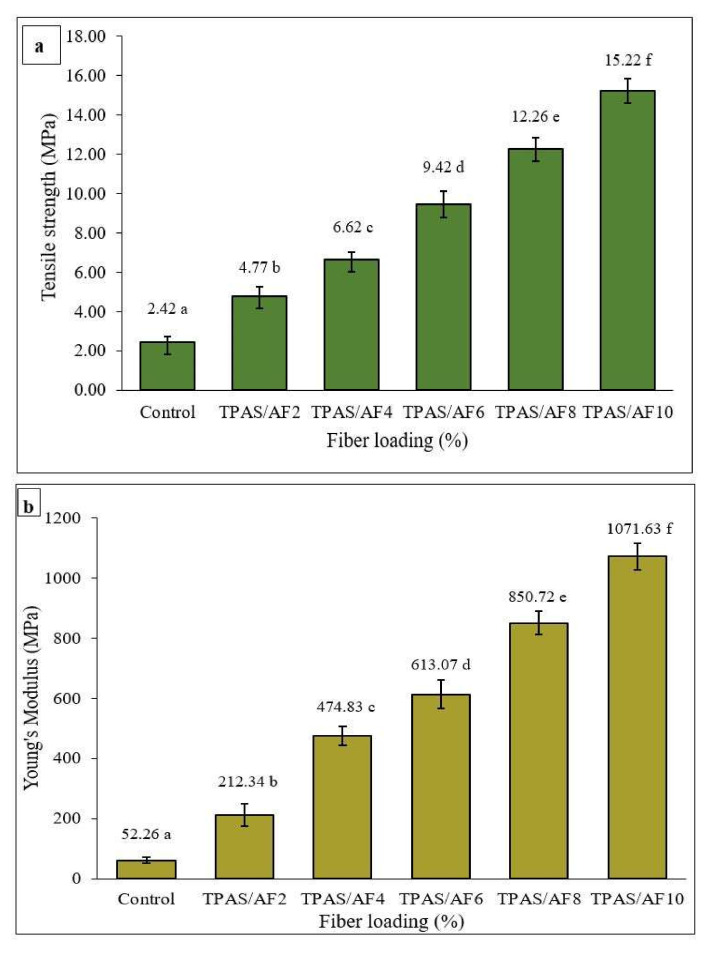
Mechanical performances of the thermoplastic composite film. (**a**) TS; (**b**) YM; (**c**) EAB; and (**d**) tear strength and load curve. Values with different letters in the figures are significantly different (*p* < 0.05).

**Figure 6 polymers-14-00388-f006:**
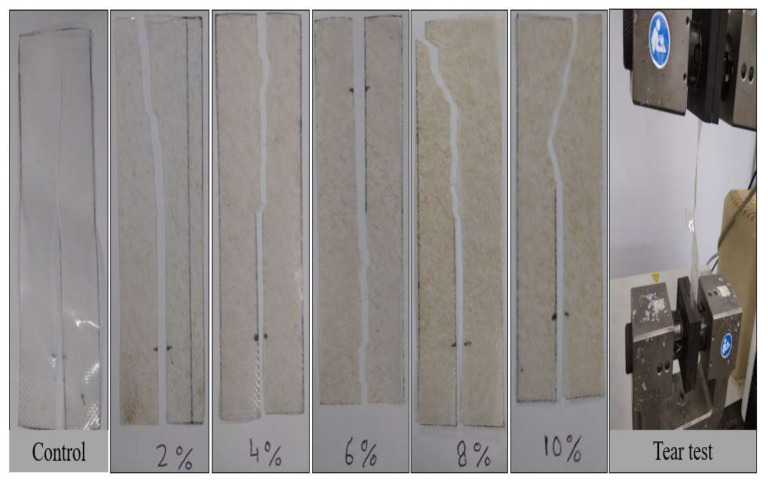
Tested specimens after the tear test of control and composite films with different fiber loadings.

**Figure 7 polymers-14-00388-f007:**
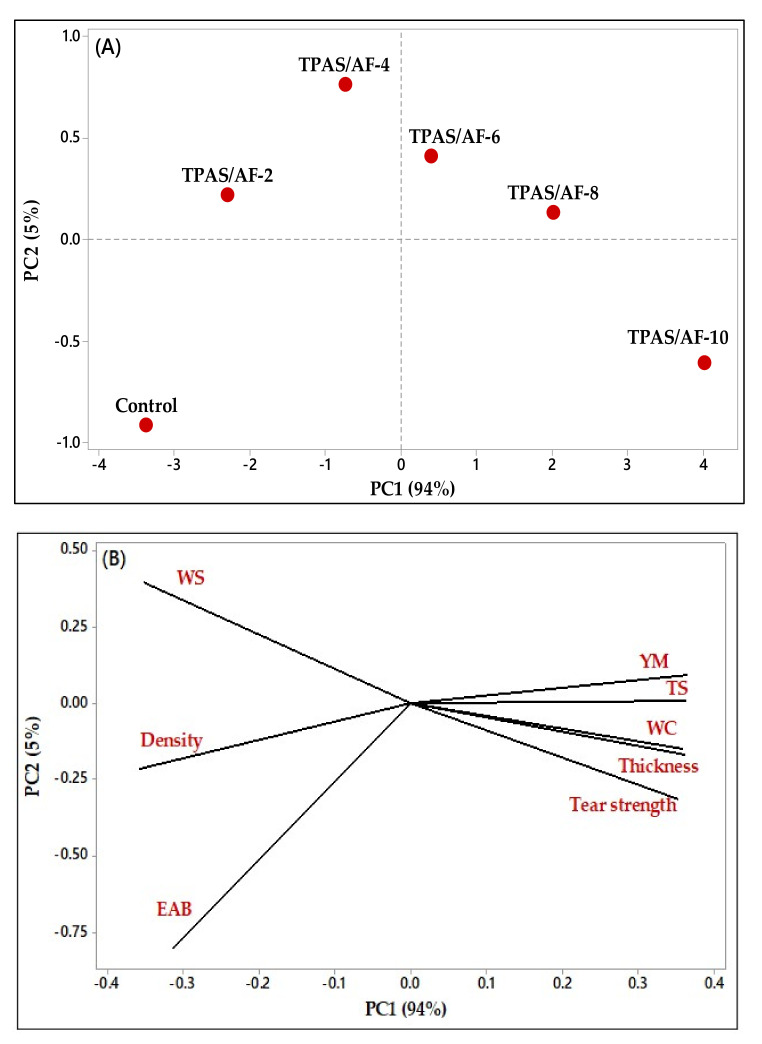
Plotting of the observations related to the physical and mechanical properties data of the variables (Control TPAS, TPAS/AF-2, TPAS/AF-4, TPAS/AF-6, TPAS/AF-8, and TPAS/AF-10) loaded on the multi-dimensional space of PCA as principal component 1 and 2 (PC1 and PC2). Specifically, 94% on PC1 and 5% on PC2 are loaded. (**A**) the location of different fiber loading; (**B**) the location of different physical and mechanical properties.

**Table 1 polymers-14-00388-t001:** A composite formulation for control and TPAS/AF composites.

Composite	Starch (g)	Fiber Loading (%)	Glycerol (g)	Distilled Water (mL)
**Control**	5	0	1.5	100
**TPAS/AF-2**	5	2	1.5	100
**TPAS/AF-4**	5	4	1.5	100
**TPAS/AF-6**	5	6	1.5	100
**TPAS/AF-8**	5	8	1.5	100
**TPAS/AF-10**	5	10	1.5	100

**Table 2 polymers-14-00388-t002:** Physical performances of the TPAS/AF biocomposite films.

Composite	Thickness (μm)	Density (g/cm^3^)	Water Content (%)	Water Solubility (%)
**Control**	163 ± 1.53 ^a^	1.33 ± 0.02 ^c^	9.77 ± 0.54 ^a^	32.62 ± 3.98 ^a^
**TPAS/AF-2**	169 ± 1.53 ^a,b^	1.32 ± 0.01 ^c^	9.82 ± 0.20 ^a^	33.19 ± 0.51 ^d^
**TPAS/AF-4**	174 ± 1.53 ^b^	1.29 ± 0.01 ^b^	10.51 ± 0.41 ^a,b^	31.76 ± 1.06 ^c,d^
**TPAS/AF-6**	181 ± 1.00 ^c^	1.28 ± 0.02 ^b^	11.19 ± 0.67 ^b,c^	29.16 ± 0.53 ^b,c^
**TPAS/AF-8**	186 ± 5.77 ^c^	1.26 ± 0.02 ^a,b^	11.65 ± 0.17 ^c^	26.29 ± 0.46 ^b^
**TPAS/AF-10**	203 ± 5.77 ^d^	1.25 ± 0.03 ^a^	12.71 ± 0.54 ^d^	22.56 ± 0.44 ^a^

Data are expressed as the mean value of replication (*n*) ± SD; for the same column, the different letter indicates a significant difference (*p* < 0.05).

**Table 3 polymers-14-00388-t003:** Effect of different environmental conditions on water absorption of TPAS/AF composite films.

Composite	Water Absorption (%)								
**Types of Water**	**Distilled Water**			**Seawater**			**Rainwater**		
**Time**	**30 min**	**60 min**	**180 min**	**30 min**	**60 min**	**180 min**	**30 min**	**60 min**	**180 min**
**Control**	166.2 ± 4.1 ^c^	175.5 ± 5.8 ^c^	189.6 ± 7.5 ^c^	36.4 ± 0.8 ^a^	40.9 ± 0.8 ^a^	48.5 ± 0.5 ^a^	31.8 ± 0.8 ^a^	38.8 ± 0.5 ^a^	40.9± 0.8 ^a^
**TPAS/AF-2**	133.7 ± 1.4 ^a^	149.6 ± 1.4 ^a^	171.7 ± 0.5 ^a^	120.0 ± 2.2 ^d^	128.1 ± 2.5 ^d^	140.0 ± 0.8 ^d^	108.7 ± 0.8 ^b^	121.4 ± 0.8 ^b, c^	127.3 ± 0.9 ^b^
**TPAS/AF-4**	111.1 ± 4.9 ^b^	117.8 ± 5.7 ^b^	133.3 ± 5.7 ^b^	110.4 ± 0.8 ^b^	115.2 ± 0.5 ^b^	120.9 ± 0.9 ^b^	107.1 ± 0.8 ^b^	117.1 ± 1.2 ^b^	125.2 ± 1.2 ^b^
**TPAS/AF-6**	94.4 ± 1.6 ^b^	100.0 ± 2.9 ^b^	112.7 ± 4.1 ^b^	138.9 ± 0.8 ^d^	150.6 ± 1.6 ^d^	157.6 ± 0.8 ^d^	62.7 ± 5.6 ^d^	138.5 ± 6.4 ^d^	148.0 ± 5.4 ^d^
**TPAS/AF-8**	89.8 ± 5.7 ^b^	95.2 ± 7.4 ^b^	110.2 ± 7.7 ^b^	104.6 ± 2.9 ^c^	115.0 ± 4.0 ^c^	120.6 ± 3.6 ^c^	96.6 ± 2.2 ^b^	101.4 ± 2.2 ^b^	107.0 ± 1.7 ^b^
**TPAS/AF-10**	91.2 ± 2.5 ^b^	92.3 ± 2.9 ^b^	100.0 ± 4.1 ^b^	100.1 ± 3.1 ^d^	108.6 ± 2.2 ^d^	117.7 ± 3.1 ^d^	96.5 ± 7.5 ^c^	105.0 ± 8.2 ^c^	112.8 ± 8.4 ^c^

Data are expressed as the mean value of replication (*n*) ± SD; for the same column, the different letter indicates a significant difference (*p* < 0.05).

**Table 4 polymers-14-00388-t004:** Summary of the analysis of variance (ANOVA) of mechanical properties.

Variables	df	Tensile Strength	Young Modulus	Elongation at Break	Tear Strength
**Mixture**	4	0.00 *	0.00 *	0.00 *	0.00 *

* Note: Significant difference at *p* ≤ 0.05.

## Data Availability

The data that support the findings of this study are available from the corresponding author, upon reasonable request.

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
