# Peer review of "Physical, Mechanical, and Morphological Performances of Arrowroot (Maranta arundinacea) Fiber Reinforced Arrowroot Starch Biopolymer Composites"

_polymers, 2022, doi:10.3390/polym14030388_

Round 1
Reviewer 1 Report
The authors proposed the fabrication of biopolymer composite films of TPAS/AF and their vital characterizations for the food packaging application. The manuscript was very well organized. The discussion was also reasonable with a lot of comparison to other previous works. I recommend to publish with Polymers after revising these following points.
- Abstract: “The tensile and tear strengths of TPAS/AF composites were increased significantly from…” It would be better to add the basis of comparison. For example, “…the tensile strengths of TPAS/AF was increased significantly from 2.42 to 15.22 MPa as compared to the control TPAS film…”
- Section 2.2, Line 2: Please specify the full name of the abbreviation “AS”. Is it the same with “TPAS”? It would be better to use only one for easy understanding by the readers.
- Table 1, Table 2, Table 3: The font, size, and the style of table should be followed the template of MDPI-Polymers.
- Section 2.3.2, Equation 1: The presence of equation was very confused and easy for misunderstand. The unit (g/cm3) should be separated from the equation and write down in the sentence instead. For example, “…m and v are the weight (g) and volume (cm3) of the film sample,…”
- Section 3.1: For the difference of thickness of films, is it depend on the technique of film casting? How to make sure that the film casting process was well controlled and done with the same methods for all samples?
- Section 3.2-3.4: For TPAS/AF-10, the water content was increased; however, the water solubility was decreased as compare to the control. In addition, the water absorption was also decreased. Do you have any discussion about the relationship between water content, water solubility, and water absorption of TPAS/AF-10?
- Table 2: It would be better to explain about the meaning and the difference of the small letter “a”, “b”, “c”, “d” and “p<0.05”.
- Section 3.4, Figure 2, Table 3: It can be found that the absorption of rainwater was similar to seawater. Do you have any explanation about the effect from rainwater on films?
- Section 3.4, at the second paragraph: It would be better to cite the reference of “Fick’s law”.
- Figure 2, at the figure caption: It would be better to cite the reference of “Duncan’s test”.
- Section 3.5, the title: The “*” at the title of section should be deleted.
- Section 3.5, Figure 3: From Figure 3, it can be found that the peak at 1367 cm-1 (C-H) was increased after loading of fiber more than 4 wt%. Do you have any explanation about this issue? In addition, the explanation of 1337 cm-1 (-OH) in Section 3.5 was unclear. It is easy confusing between 1337 cm-1 in Section 3.5 and 1367 cm-1 in Figure 3.
- Section 3.6, Line no. 7: “…the cross-section in the surface of control film samples shown in Figure 4(A)…” For the cross-section image, should it be “Figure 4(G)” instead of “4(A)”?
- Section 3.6, Line no. 13: Should it be “Figure 4(H, I J)”?
- Section 3.6, Line no. 15: Should it be “Figure 4(K and L)”?
- Section 3.7, Line no. 1: Should it be “Figure 5(a, b, c)”?
- Section 3.8, Figure 5(d): It would be better to explain about the load curve on fiber loading%.
- Section 3.8, Figure 6: The explanation of the visual investigation of the direction of tear propagation of each films was unclear.
Reviewer 2 Report
1. The abstract should mention the significance of the work clearly.A line needs to be added addressing why the biopolymer composite film is important in the food packaging industry.
2. Line 57,58 – “There are also some drawbacks….reinforcements.”This line is confusing because the authors start with drawbacks but mentions only the benefit. This sentence needs clarification.
3. Tables 2,3 and Figure 2- Please clarify what are a,b,c,d.
Reviewer 3 Report
The title of this study is: Physical, mechanical and morphological performances of arrowroot (Maranta arundinacea) fibre reinforced arrowroot starch biopolymer composites. In this study aimed using glycerol as a plasticizer on the characteristics of biopolymer film and arrowroot bagasse fiber (2, 4, 6, 8, and 10 wt. %) as the reinforcement.
I commented on the manuscript and the comments are presented below:
Part 1: Introduction.
The Introduction to the study is too broad and does not end with a clearly stated p urpose or goals that the Authors wish to pursue. This should be changed. I suggest supplementing the Chapter with additional information related to other new methods and devices in studied research, for example:
“Characteristics of Newly Developed Extruded Products Supplemented with Plants in a Form of Microwave-Expanded Snacks”.
“Optimisation of extrusion variables for the production of corn snack products enriched with defatted hemp cake”.
Part 2: Material and Methods
Only the basic statistical analysis was used to describe the differences. Have the Authors attempted to use other more comprehensive statistical analyzes, e.g. principal components analysis of PCA? With such a large number of parameters tested, which may affect the characteristics examined, the Principal Component Analysis (PCA) should be used to results analyzed. More advanced statistical analysis should be performed. The use of advanced statistical methods to fully describe the relationship between the parameters studied and the aspects of the research work carried out in the presented manuscript. You can determine the strength of the influence of a particular parameter on the variance of the system. At the same time, correlation relationships between the determined parameters can be determined.
Part: 3 Results and discussion
For the most part the Results section is well structured.
In the Discussion chapter, there is no full comparison and confrontation with the research of other authors in this area. The results were not fully discussed. A full discussion of the results obtained with other work in this field should be carried out in more aspects. I suggest supplementing the Chapter with additional information, for example:
“Physical Properties, Spectroscopic, Microscopic, X-Ray, and Chemometric Analysis of Starch Films Enriched with Selected Functional Additives”.
“Effects of extrusion conditions and nitrogen injection on physical, mechanical, and microstructural properties of red lentil puffed snacks”.
“Effects of selected process parameters on expansion and mechanical properties of wheat flour and whole cornmeal extrudates”.
“Effect of Supplementation of Flour with Fruit Fiber on the Volatile Compound Profile in Bread”.
“Use of principal component analysis (PCA) and hierarchical cluster analysis (HCA) for multivariate association between bioactive compounds and functional properties in foods: A critical perspective”
Part: 4 Conclusion
The Conclusions chapter contains information obtained after conducting experiments but performing only base statistical analyzes and were no comparison and confrontation with the research of other authors in this area.
Part: References.
The literature used is appropriate but should be supplementing about the items from the last years of publication about similar problem.
Round 2
Reviewer 1 Report
Thank you very much for the revised manuscript. The contents of this version were greatly improved. However, some questionable points have been remained.
- It seems like the authors changed the word of “fibre” to “fiber” especially in the abstract. How about the title of manuscript? It would be better to change to …“fiber” reinforced arrowroot…
- About the rainwater and seawater absorption (Section 3.4, Figure 2, Table 3); From Figure 2 (B, C) and Table 3 at 180 min of TPAS/AF6, 8, 10, it can be found that the absorption of rainwater was higher than seawater. (Rainwater: 158, 121, 118%; Seawater: 148, 107, 113%). This result was in opposite with the explanation in Section 3.4 (Line no. 360-363). Please recheck about the results, the explanation, and the comparison between the effect from rainwater and seawater absorption on TPAS/AF. Why the absorption of rainwater was higher than seawater even rainwater had lower in density and salinity compared to seawater?
- Some error in English writing have been remained. For example, at Line no. 513; “Afs concentartion” -> “AFs concentration”.
Reviewer 3 Report
The authors referred to the comments from the previous review for the manuscript titled: Physical, mechanical, and morphological performances of arrowroot (Maranta arundinacea) fibre reinforced arrowroot starch biopolymer composites. I accept explanations. In the future, I suggest using more precise describing relationships between the parameters studied. They supplemented the discussion and strengthens the message and importance of information in the manuscript.
Round 3
Reviewer 1 Report
Thank you very much for the revised version. I accept for publication with Polymers in this present form.